# “I Needed to Know, No Matter What I Do, I Won’t Make It Worse”—Expectations and Experiences of Last Aid Course Participants in Germany—A Qualitative Pilot Study

**DOI:** 10.3390/healthcare11040592

**Published:** 2023-02-16

**Authors:** Chantal Giehl, Nino Chikhradze, Georg Bollig, Horst Christian Vollmar, Ina Otte

**Affiliations:** 1Institute of General Practice and Family Medicine (AM RUB), Medical Faculty, Ruhr University Bochum, 44801 Bochum, Germany; 2Department of Anesthesiology, Intensive Care, Palliative Medicine and Pain Therapy, HELIOS Klinikum, 24837 Schleswig, Germany; 3Last Aid Research Group International (LARGI), 24837 Schleswig, Germany

**Keywords:** Last Aid course, palliative care, caring relatives, qualitative research, interview, Germany

## Abstract

Background: The need to care for a loved one in a palliative state can lead to severe physical as well as psychological stress. In this context, Last Aid courses have been developed to support caring for relatives and to stimulate the public discussion on death and dying. The purpose of our pilot study is to gain an understanding of the attitudes, values, and difficulties of relatives caring for a terminally ill person. Methods: A qualitative approach was used in form of five semi-structured guided pilot interviews with lay persons who recently attended a Last Aid course. The transcripts of the interviews were analyzed following Kuckartz’s content analysis. Results: Overall, the interviewed participants have a positive attitude toward Last Aid courses. They perceive the courses as helpful as they provide knowledge, guidance, and recommendations of action for concrete palliative situations. Eight main topics emerged during analysis: expectations regarding the course, transfer of knowledge, reducing fear, the Last Aid course as a safe space, support from others, empowerment and strengthening of own skills, and the improvement needs of the course. Conclusions: In addition to the expectations before participation and the knowledge transfer during the course, the resulting implications for its application are also of great interest. The pilot interviews show initial indications that the impact, as well as supportive and challenging factors regarding the ability to care for relatives to cope, should be explored in further research.

## 1. Introduction

Dying and death are part of life and determine it in many ways. Most people want to be cared for in their familiar surroundings for as long as possible and ultimately die there [1]. In 2019, 70% of the people in Germany expressed the wish to die at home [2]. Good outpatient palliative care can meet this wish of patients and avoid hospitalization and cost-intensive therapies; at the same time, it can lead to a higher quality of life and better symptom control at the end of life [3]. Simultaneously, a shortage of qualified professionals poses a major challenge to the German healthcare system to ensure adequate and high-quality outpatient palliative care [4]. Palliative care is covered by a nursing care insurance which is called “Pflegeversicherung”. It was introduced in Germany in 1995 and is mandatory for members of state and private health insurance schemes. The costs are borne jointly by employees and employers. The “Pflegeversicherung” covers the costs of long-term nursing care for the old or disabled. It also pays a certain amount to those who care for relatives at home and reimburses their contributions to social security.

Family caregivers provide the majority of all care services in palliative care and are significantly involved in the outpatient care of palliative patients [5,6]. Kellehear’s 95% rule underlines this point. It states that palliative patients spend most of their time with their families and only 5% of their time with healthcare professionals [7]. In this context, family carers in outpatient palliative care in Germany are often exposed to a large number of critical health events when caring for their terminally ill family members. These can cause both physical and psychological stress, such that they require support needs [4,8,9,10,11,12]. This distress is associated with levels of anxiety and helplessness, as well as deterioration in their overall health [9]. For this reason, many of these people do not feel competent enough to care for dying relatives at home. In addition, they often lack the necessary basic knowledge of palliative care and end-of-life care [13]. Good outpatient palliative care includes family education [14] and especially early training in caring for relatives. In this context, it is crucial to support family caregivers in their knowledge and skill development [4,11,15].

In 2014, Last Aid courses were developed in response to this outlined need. Last Aid courses aim to provide the general population with knowledge about death, dying, and palliative care. They are intended to equip family carers and all interested citizens with the necessary basic knowledge about caring for the seriously ill and dying, and thus to alleviate the fears of the course participants through low-threshold access to information. An improvement of the situation for caring for relatives, but above all for the terminally ill and dying, is to be achieved. The courses are offered at various locations by course instructors who have previously been trained by Letzte Hilfe Deutschland [16]. Last Aid courses are usually held by two certified instructors with experience in the field of palliative care (e.g., physicians, nurses, hospice volunteers, priests, and social workers). The courses are offered in different community settings, such as hospitals, churches, schools, and other public places. They are also offered as part of the education of healthcare workers, policemen, or the military. The course (four modules of 45 min each, a total duration of 3.5 to 4 h, including breaks) offers participants not only the opportunity to acquire knowledge but also to reflect on their own experiences with the topic of dying and death and to obtain a palliative attitude [13].

By 2019, more than 26,000 people in German-speaking countries had already attended a Last Aid course [17]. The number of participants has continued to increase since then. At present, more than 50,000 people have participated in Last Aid courses in Germany, Austria, and Switzerland. Participants are recruited via posters and information in the workplace, newspapers, radio, television, and the internet. The course is offered by different organizations such as the Red Cross, hospices, hospitals, churches, and schools, among others. Therefore, it is indispensable to evaluate the course curricula and to further research participants’ experiences. To date, there is limited scientific data on the participants and the effects of the courses offered, as no comprehensive scientific evaluation has yet been conducted. This is exactly the starting point for the article presented here, which proposes the first results of the pilot interviews with course participants as a feasibility study for our main research project. So, it is the aim of this study to optimize and complement the interview topic guide for the main research project.

## 2. Materials and Methods

A qualitative design has been chosen for the pilot study presented here. It consists of five qualitative, semi-structured interviews, which are particularly suited to understanding lay people’s attitudes, values, and difficulties in caring for a terminally ill person. After a successful pilot, a larger main study will be focusing on barriers and benefits for laypersons when participating in a Last Aid course.

### 2.1. Sampling and Data Collection

The purposive sampling consisted of five course participants who indicated an interest in the pilot study. After the study, information had been sent to the course instructors, and they forwarded it to the course participants. The course participants reached out to the course instructors on their own and expressed their interest in participating in the study. In response, the course instructors contacted GB. Chosen course participants were contacted via e-mail or telephone outlining the research. Semi-structured telephone interviews (one via video call using Zoom per request), approximately 25 min in length, were conducted by C.G. The participants were interviewed individually. The participants are all female and lived in Germany and Denmark. Due to the small sample size, we are not able to disclose further characteristics, as their anonymity would be affected. The interviews took place between November 2021 and February 2022. A preliminary semi-structured interview topic guide was used for all interviews, which evolved as new insights were gained during the data-gathering process. The original pilot interview topic guide is in German, as the interviews were conducted in German. For comprehensibility of the methodology, the English translation can be found in the Appendix A. The existing interview topic guide can be revised and adapted for the larger project. The revision and adaptation of the interview topic guide led to a more in-depth exploration of the topics addressed in the aim of the study in preparation for the main project. Question sets included the most important challenges when administering palliative care, participants’ takeaways from Last Aid courses, and reasons for participating in Last Aid courses, as well as suggestions for improvement. All participants provided verbal and written informed consent.

### 2.2. Analysis

The interviews were transcribed verbatim. The transcripts were reviewed independently by IO and CG. Both IO (sociologist and medical ethicist) and CG (health services researcher) are specialized in qualitative research methods and interviewing techniques. CG carried out an independent analysis of all transcripts using MAXQDA software while following Kuckartz’s steps of content analysis [18]. In the first step, the data was coded separately, moving from concrete passages to more abstract levels of coding, including emerging themes. C.G. and I.O. then discussed the codes and C.G. re-coded potentially unclear passages again. Critical reviews and plausibility checks of each analysis of each interview were performed in order to help us to become aware of our own backgrounds and potential bias (reflexivity) [19]. The codings were then reviewed by a third independent researcher to ensure inter-rater reliability. After two interviews, a preliminary coding guide was developed which was adapted continuously throughout the analysis, adding new codes emerging from the material, if necessary. In research group meetings, all findings were critically tested and discussed. In this way, any discrepancies could be resolved.

For this paper, C.G. translated the quotes and back-translated them to eliminate any confusion of meaning [20].

## 3. Results

The presented results consist of the eight major topics that emerged during the analysis of the interviews. They focus on the expectations and experiences of the course participants and give an initial overview of and insight into these topics.

In general, the interviewed course participants have a positive attitude toward Last Aid courses. They perceive the courses as helpful as they provide knowledge and methods of application for concrete palliative situations. In addition to expectations prior to participation and knowledge transfer during the course, the resulting implications regarding the application are of great interest as well.

Therefore, the results are clustered and presented as follows: expectations regarding the course, transfer of knowledge, reducing fear, Last Aid courses as a safe space, support from others, empowerment and strengthening of own skills, and improvement needs of the course. For a better overview of the topics, see the following figure (Figure 1).

### 3.1. Expectations Regarding the Course

Attending a Last Aid course is based on an underlying motivation, which is tied to certain expectations of the course. The motivation arises from either interest in the topic or a concrete palliative situation, for example at home or with a close circle of friends and family. However, people might also go into the course with no expectations, as they do not know exactly what to expect in terms of content:

“*Actually, I didn’t have any great expectations. I just thought, “Well, it can’t hurt!”. And somehow I find the topic interesting anyway.*”[Interview Participant 4]

“*I didn’t really have any expectations. I went into the course completely free. I let myself be surprised, I couldn’t imagine […] how it would work or what I would actually have to face. I said: I’m interested, but I don’t know what to expect.*”[Interview Participant 1]

Other people might have a concrete idea of what they want to take away from the course due to a concrete palliative situation and their resulting need for support and assistance:

“*Because it was already clear that my friend was already in the hospice and I had the feeling that I still needed some kind of support,* i.e., *guidelines on how to interact with her. […] Well, I somehow lacked the know-how, I’ll say. And that was the first time that I accompanied someone so closely.*”[Interview Participant 3]

“*Actually, the expectation was that you would actually receive assistance. In other words, concrete assistance. No? Because when you care for someone at home, it’s something different than when they’re in a nursing home, so to speak.*”[Interview Participant 5]

### 3.2. Transfer of Knowledge

Participants hoped for said assistance mainly through the transfer of knowledge. One main component was information about physical changes in the process of dying. These were particularly essential for one interview participant:

“*Especially helped [was to know] how to make it a little easier for the sick person and what’s going on physically in a dying person. Because you actually have no idea what happens in the body when it dies. And, yes, she explained that very, yes, more or less vividly and, yes, I actually took away quite a lot.*”[Interview Participant 1]

While knowledge transfer is an essential point of Last Aid courses, the implementation of it and how the newly gained knowledge equips the course participants for future palliative care situations were as equally important. Through the knowledge gained, course participants were strengthened in their daily practices. After attending the course, they report approaching palliative situations in a more calm manner, as they now have a more profound background knowledge which gives them security and reduces their helplessness:

“*Yes, that you simply approach the situation in a calmer and more relaxed way.*”[Interview Participant 2]

“*However, I would actually recommend such a course to anyone who has parents or parents-in-law in the house and who wants to and can take over care [...] because you simply [...] have a slightly different approach than if you have no idea at all [...] what you’re facing. You know that everyone dies one day, but, yes, when the time comes, you are not so helpless. And that’s a little bit of security, that there’s someone there when you don’t know what to do, at least just to ask questions, or who comes by once in a while and maybe comforts you.*”[Interview Participant 1]

### 3.3. Reducing Fear

Through the transfer of knowledge and the open discussion in the courses, the fears of participants were reduced and safety in everyday life was promoted. Since the topics of dying, death, and grief are rarely reflected upon in society, there is a lack of knowledge about the last phase of life. This is why the fears of informal caregivers in dealing with palliative situations are present:

“*What has actually remained is an overall impression and a feeling that this is a topic you don’t have to be afraid of, where you should simply deal with it in a lighthearted manner. [...] I think the most important thing is that the fear of dealing with it at all was taken away. [...] In any case, I’m no longer afraid of it. And I think that is also one of the most important things, yes, that you simply lose this fear.*”[Interview Participant 4]

### 3.4. The Last Aid Course as a Safe Space

Not only the content but also the environment in which a Last Aid course takes place is essential. Only in a safe environment can participants fully open up to talk and learn about often very emotional topics that are otherwise rarely addressed in daily life. Therefore, it is important to fulfill certain conditions, such as building and offering a safe space in which the participants can trust each other and the course instructor and also be emotionally supported:

“*First of all, it was incredibly relieving to sit between people who had the same reason, so to speak, and the same topic and questions and also to be able to allow such a sadness. [...] Well, it was such a protected space, it was built in a way that you could just cry there. Because I was always strong, strong, strong, with the family, with friends, and somehow pushed it far away. And there was room for it. I actually found that really moving.*”[Interview Participant 3]

“*There were also very personal things told and you also had the feeling that you can tell everything there and that it stays within the group.*”[Interview Participant 4]

### 3.5. Support from Others

Support from others includes, on the one hand, professional support from care providers, such as nursing services. On the other hand, the exchange with the other course participants can also offer support, as they are in similar situations with similar challenges and often similar emotions.

#### 3.5.1. Support from Care Providers

Although the Last Aid course provides basic palliative care knowledge for the general population, it cannot and is not intended to replace palliative care provided by professional caregivers. The interviewed course participants experienced that through their participation they got to know care providers they could contact afterward. This gives them a certain security because they know who they can turn to when the limits of their own knowledge or their own possibilities for action have been reached:

“*Well, so what helped me most was actually the security of knowing that I wasn’t completely on my own in the situation and that I knew where to get help.*”[Interview Participant 2]

The same participant reported calling the course instructor, who is also a palliative care nurse, at night for reassurance about medication:

“*Then I was able to call [a palliative care nurse] and she told me, yes, do that, do that, do that and then you also have a good feeling that you’re not doing anything wrong.*”[Interview Participant 2]

#### 3.5.2. Support from Other Course Participants

The support from other course participants is expressed mainly through their similar situations with similar challenges. Meeting other people affected by the condition and exchanging experiences with them can have a supportive effect. In addition, participants can give each other tips and support:

“*So I do believe that it helps at that moment. So, when you sit with others who are also in that position. And say, man, no, with us we do it like this or like this. Right? And it’s difficult anyway to put everyone who dies into one category, so to speak.*”[Interview Participant 5]

### 3.6. Empowerment and Strengthening of Own Skills

The aspects described above directly and indirectly lead to empowerment and strengthening of the skills of the participants of a Last Aid course. Through the knowledge gained and the resulting confidence, course participants feel empowered and are encouraged to address challenges in a context that they would not have dared to address before:

“*The day she died, I was so strengthened by the course to tell the truth that I managed to pick them all up. I absolutely organized it and ran it so it didn’t turn into chaos. [...] I was pretty proud of myself. So the only time I allowed myself to do that was when I was at home with my husband at night, really crying and saying, today she died.*”[Interview Participant 3]

“*But I just had the [person] in the back of my mind and everything you heard in the Last Aid course, and then we said: Let’s at least try it.*”[Interview Participant 2]

Strengthening self-confidence is another effect that can result indirectly from participation. Through the exchange with course instructors and other course participants, the participants reported acting more self-confidently and, above all, that listening to one’s intuition is often a good way to go:

“*For me, the course was actually, if I have to draw a conclusion, a confirmation that no matter how you do it, you do it right. So that you shouldn’t blame yourself afterwards for not having done this or that. Everyone does it as well as they can. And I found that very comforting for myself.*”[Interview Participant 4]

### 3.7. Improvement Needs of the Course

Although many positive aspects were mentioned regarding the impact of the courses, participants also suggested some room for improvement. Above all, the participants would like more information about local structures and professional networks. Even if you do not need it at that moment, it is helpful to come back to it later if needed. Furthermore, more advertising should be conducted to spread the courses further among the general audience. Various media channels should be used for this purpose:

“*Maybe just more advertising, like that. Even more publicity. [...] I think it would be very good if the same happens even more in the media.*”[Interview Participant 3]

## 4. Discussion

Although identified through a small number of respondents, the results of the pilot interviews show that the Last Aid course seems to be an appropriate means to prepare relatives for outpatient palliative care, as it increases people’s confidence and empowerment. Sample determination was not used based on data saturation. This is a limitation of the results presented here. Further research would be needed to confirm our findings.

However, it was very challenging for the interview participants to talk about their concrete expectations of the Last Aid course. It is known from the literature that people are not prepared for the different challenges that arise within the family due to a terminal illness. Therefore, relatives tend to feel overwhelmed and often do not know how to act in different demanding situations [4,10,11,12]. This results from the uncertainty about the course of the disease or how an incurable disease manifests itself and how to appropriately deal with the symptoms. Understandably, this can result in confusion and anxiety. Relatives, therefore, often know their need for information but cannot specify exactly what that information should be.

A study by Dillen et al. aimed to identify the components that contribute to an increase in the feeling of safety within the context of palliative care. A total of 197 patients and 10 caregivers (relatives) were interviewed [14]. The feeling of safety was primarily related to the availability and provision of information and education, professional competence, patient empowerment, and trust. These phenomena coincide with the pilot interviews presented here. Through the contents learned, course participants were able to experience security, reduce fears, and strengthen their skills in dealing with seriously ill patients. Gaining security in everyday life and reducing fears are important factors in palliative care because it allows informal caregivers to feel safe providing home care during the last phase of a loved one’s life [6,12] and thus reduces their worries in everyday life and about the ill person [5,11]. This could also reduce their tendency to physical as well as psychological stress [5,10] through increased self-confidence [21,22].

According to a study by Hacker, Slobodenka, and Titzer, a closed and safe space ensures the well-being of course participants [23]. It allows people to talk to each other about their challenges and to benefit from each other’s experiences as well as from the theoretical course content. These results were also confirmed in the pilot interviews, as the participants perceived the Last Aid course as such a space and were, therefore, able to speak openly about their everyday family situations and the challenges they face. One example of the open exchange is the encouragement that rather unconventional approaches can be just right for the person. These include, for example, taking another vacation or moving to a nicer hospice room.

Support from others in similar situations was perceived by the interviewees to be especially helpful and accommodating. During a terminal illness, carers’ daily routines are reduced. The demands of everyday life and illness are manifold. In this situation, many family members often overlook the stresses within the family, be it their stresses or the stresses of others. Concerns about the well-being of the seriously ill person and maintaining daily routines overshadow all other needs. Therefore, it is important to create a different, external perspective on the different dynamics of the disease, to recognize resulting demands within a family system, and to support the caregiver or relative in the process of coping [8,10,11,24].

Nevertheless, the interviewed course participants also need theoretical content and practical support in everyday life that is structured according to general and individual needs, which is in line with the results of the literature. In the last phase of life of a close person with a terminal illness, people have to deal with the changing complex situations in everyday life [11]. These are characterized by the fact that family members are often overwhelmed due to their multiple roles as a child, (spousal) partner, carer, and friend, as well as an unclear picture of how to care for their family member. This makes it difficult for them to deal with their relatives in the last phase of life and to take over care at home [12]. Relatives also have to deal with the unpredictable course of the disease and need close support from health professionals in everyday life [11]. In this context, the negotiation of practical actions between health professionals and informal caregivers serves to support them [24]. Thus, health literacy and empowerment of relatives can be promoted as early as possible, and the specific consideration of relatives’ needs can ensure home-based care in the long term [25,26,27]. This would allow the family to accommodate the wishes of the dying person: to care for them in their familiar surroundings for as long as possible and ultimately to die there [1,28,29,30]. Similar results were confirmed in the pilot interviews, as the participants wish to have concrete contact persons whom they can get in touch with in cases of emergency or if they have further questions about their specific situation. These contact persons may include general practitioners, nurses from a home care service, specialized palliative care teams, and others. Information about the local structures would be helpful, as would a handout with contact details of the relevant institutions and contacts of support.

## 5. Conclusions

Overall, by conducting pilot interviews, initial indications of categories were obtained that provide valuable guidance for adapting the interview topic guide and serve as the basis for the main study. The findings of this pilot study indicate impacts beyond just knowledge transfer, i.e., empowerment, creating a safe space for learning, strengthening skills, and managing anxiety. Thus, the interviewed course participants showed a positive attitude toward Last Aid courses. They found the courses helpful because they gained knowledge and methods to use in concrete emergencies and were able to practice them afterward. In addition to the expectations before participation and the knowledge transfer during the course, the resulting implications are also of great interest. Since the number of initial interviews is manageable, further research is needed to answer the questions discovered through this pilot research and to achieve data saturation. Parallel to the interviews with the course participants, another perspective, namely that of the Last Aid course instructors, should also be explored to evaluate Last Aid courses and their effects in more depth in order to provide a richer picture. The inhibiting, as well as facilitating, factors for the participants and lecturers should also be investigated—especially since new web-based formats (online or hybrid) for the courses were established due to the COVID-19 pandemic [31] and could be used more frequently in the future. To explore the new course formats and the other phenomena mentioned above, the initiation of a more extensive academic project is essential.

## Figures and Tables

**Figure 1 healthcare-11-00592-f001:**
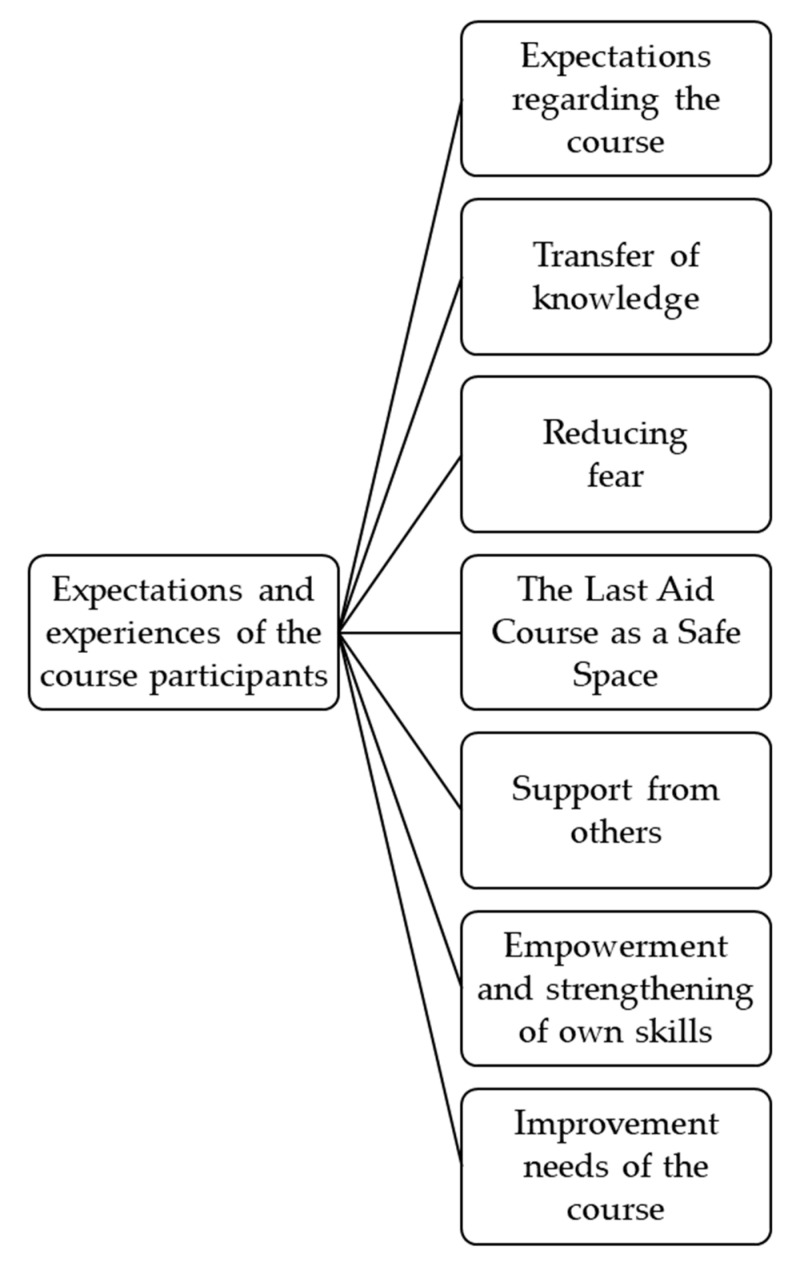
Main topics emerged from the analysis.

## Data Availability

The data presented in this pilot study are available upon request from the corresponding author. The data are not publicly available.

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
