# Peer review of "“I Needed to Know, No Matter What I Do, I Won’t Make It Worse”—Expectations and Experiences of Last Aid Course Participants in Germany—A Qualitative Pilot Study"

_healthcare, 2023, doi:10.3390/healthcare11040592_

Round 1

Reviewer 1 Report

This research is interesting and could have a great impact on the evaluation of the courses applied to family members, both for the benefits they can offer for care and for their own feelings (overburden, guilt, helplessness, feeling of not doing the right things or good things, etc). However, there are some considerations that make this article need to be improved.

First, the study sample. Reading the article it is clear that data saturation has not been reached, firstly because not all the guiding questions are answered, secondly because if 26,000 people have taken this course, a sample of 5 is excessively small and,  in Third, because you say so in the discussion.

Second, it is said to be a pilot study. I don't think this is what is called for in this article. The investigation is alive. A study starts with a pilot study, but if it is to be published in a quality journal, the research must continue and show powerful data, not just approximations.

Third, the sample is not described. An explanatory table of the characteristics of the sample is missing: Relationship, age, sex, etc.

For all these reasons, I believe that in its current state, the article does not meet rigorous research standards.

Author Response

Dear Reviewer,

We are grateful for the thorough review process; the comments were very helpful and increased the quality of our paper. Please see the attachement with a point-to-point list of replies. Substantial changes in the manuscript are marked via track changes.

We hope that the revised version of the manuscript will now be suitable for publication.

Thank you for your time and consideration.

Yours sincerely,

Chantal Giehl on behalf of the authors

Reviewer 2 Report

I appreciate this well-written description of your research.  I feel the sample size is very small, even for a qualitative study.  I would like more contextual information about the Last Aid Course, e.g. its history, the intended audience(s), and its uptake.  Do you have information as to how many participants in the course generally are actively engaged in caring for a loved one, versus those who take it for other reasons?  In addition, for US publication, it would be helpful to understand features such as the reach of home-based palliative care in Germany and its financing.  Palliative care is still confused with hospice care in the US, and their respective reimbursement systems are quite different. These factors would influence the feasibility of such a potentially valuable course offering in the US.

On that note, Bruce Jennings (2013) has advocated for the adoption of a "civic" palliative care, arguing that care for the dying should be community-based rather than family based.   In that vein, having a publicly available course such as Last Aid would certainly be beneficial.  While I am intrigued by the prospect, I fear that the existing isolation and stigma of patients with advanced illness in the US, whether in the home or or in a care facility, is difficult to overcome.  Mortality salience makes people more conservative and less expansive generally (Solomon, et al 2015).  Yet those who care for serious illness at home need all the resources they can get.

Jennings, Bruce. (2013). Chapter 16: Solidarity, Mortality: The Tolling Bell of Civic Palliative Care. In Staudt, Christina; Ellens, J. Harold (Ed.), Our Changing Journey to the End (2 volumes): Reshaping Death, Dying and Grief in America (Vol. 2, pp. 271–288). Praeger. Solomon, S., Greenberg, J., & Pyszczynski, T. (2015). The Worm at the Core: On the Role of Death in Life. Random House.

Author Response

(The authors gave the same response as above.)

Reviewer 3 Report

Providing Last Aid Course is an interesting approach in this pilot study, yet the purpose of that is somewhat unclear. The authors should provide more information about whom the course is recommend. Are the optimal participants palliative care volunteers, care providers, family members of dying persons or just whomsoever laymen? There seems to be some overlap between the “Last Aid Course” and volunteer training, albeit the Last Aid Course is very short. In addition, it would be important to get a clear view if the participants were individually interviewed or if they participated in a group interview. I don´t quite understand what means "last aid course as a safe space". I recommend that the authors should consider very carefully the optimal outcome measures: could e.g. death and dying distress scale, HADS or Beck´s inventory be used as a pre- and post-course outcome measure? Is the aim in the future to organize group teaching or individually tailored courses?

Author Response

(The authors gave the same response as above.)

Round 2

Reviewer 1 Report

No comments

Author Response

Dear Reviewer,

We are grateful for the thorough review process and for the positive feedback on the implemented review topics; the comments were very helpful and increased the quality of our paper. Please see the attachment for the point-to-point list of replies. Substantial changes in the manuscript are marked via track changes.

Reviewer 2 Report

Comments on "I needed to know, no matter what I do, I won't make it worse" 2 - expectations and experiences of Last Aid Course participants 3 in Germany - a qualitative pilot study”

I appreciate the careful attention to comments and the changes made.  I found a number of editorial problems which should not be difficult to correct.  See below:

Page 1, Line 19, 20:  “On the transcripts”: Please change “on” to “all” or whatever should be there.

Line 24: “support by others” Please change to “support from others”

Line 42: “insurance with is called” Please change to “which is called”

Page 2, line 46: “covers their social security contributions” consider changing to “reimburses their contributions to social security.”

Line 53: “cause both, physical and psychological” Please remove the comma.

Line 59: “knowledge and action development”: “action development is unclear.  Do you mean “skill development”?

Line 71: “different places”: Consider “community settings, such as”

Line 77: “have already”: Consider “had already”

Line 80: “recruited” is misspelled.

Line 86: “point of start” Consider “starting point”

Line 89: “interview guide” It is unclear what this is.

Page 3, Lines 98 and 99:  “After the study information has been sent to the course instructors, they forwareded it to the course participants.”  Should read: “After the study information had been sent to the course instructors, they forwarded it to the course participants.”

Line 109: “The existing guide”: is this the interview guide?

Line 110: “This led to a more in-depth exploration of this topic”. Please indicate what the first “this” refers to, as well as what topic “this topic” refers to.

Line 112: “participants’ take away”: Consider “take-aways”

Page 7, Line 280 “the results presented here and should be investigated in a future study”: Consider “the results presented here, and further research would be needed to confirm our findings.”

Page 8, Line 345: “This pilot findings going beyond just knowledge transfer”: Consider “The findings of this pilot study indicate impacts beyond knowledge transfer, e.g.”

Author Response

(The authors gave the same response as above.)

Reviewer 3 Report

I think the authors have done a great job with editing their ms, which has improved a lot.

Author Response

(The authors gave the same response as above.)
